# Nutrient Availability for *Lactuca sativa* Cultivated in an Amended Peatland: An Ionic Exchange Study

Jacynthe Dessureault-Rompré *, Alexis Gloutney and Jean Caron

Soil and Agri-Food Engineering Department, Laval University, 2480 Hochelaga Blvd.,
Quebec City, QC G1V 0A6, Canada; alexis.gloutney.1@ulaval.ca (A.G.); jean.caron@fsaa.ulaval.ca (J.C.)
* Correspondence: jacynthe.dessureault-rompre@fsaa.ulaval.ca

**Abstract:** Few conservation strategies have been applied to cultivated peatland. This field study over one growth cycle of *Lactuca sativa* examined the effect of plant-based, high-C/N-ratio amendments in a real farming situation on peatland. Plant Root Simulator (PRS®) probes were used directly in the field to assess the impacts of incorporating *Miscanthus x giganteus* straw and *Salix miyabeana* chips on nutrient availability for lettuce. The results showed that lettuce yield decreased by 35% in the miscanthus straw treatment and by 14% in the willow chip treatment. In addition, the nitrogen flux rate was severely reduced during crop growth (75% reduction) and the plant N uptake index was much lower in the amended treatments than in the control. The phosphorus supply rate was also significantly lower (24% reduction) in the willow treatment. The influence of sampling zone was significant as well, with most macro-nutrients being depleted in the root zone and most micro-nutrients being mobilized. Additional work is needed to optimize the proposed conservation strategy and investigate the effects of consecutive years of soil amendment on different vegetable crops and in different types of cultivated peatlands to confirm and generalize the findings of this study. Future field studies should also explore the long-term carbon dynamics under plant-based, high-C/N-ratio amendments to determine if they can offset annual C losses.

**Keywords:** nitrogen; miscanthus; willow; field experiment; lettuce; plant-based amendment; rhizosphere

## 1. Introduction

Peatlands serve a number of essential purposes, including the production of substrates and food [1]. Thanks to their high organic matter content, drained peatlands can be highly productive and suitable for agriculture, grasslands, and forestry. However, the drainage required to make organic soils suitable for agricultural activities creates conditions conducive to rapid degradation and compaction [2]. Wind and water erosion, subsidence, and soil organic matter mineralization leading to $CO_2$ losses are the main causes of the degraded soil conditions observed in cultivated peatlands [3–5]. Solutions proposed thus far have included restoring cultivated peatlands to their natural state [6–8] or changing land use drastically, to paludiculture, for example [9,10].

More recently, the use of chopped biomass crops as a soil amendment has been proposed as a way to make peatland cultivation more sustainable [11,12]. Implemented in conjunction with optimal water table management and erosion mitigation, this practice may lead to an integrated conservation management approach that would improve the long-term sustainability of cultivated histosols [12–14].

Cultivated peatland in Canada constitutes an important part of the agricultural economy. In the plain of Montreal, in southwestern Quebec, close to 12,000 ha of land is covered in deep organic soils [15]. Although small, the area plays an essential role in the production of high-value vegetable crops, such as *Daucus carota* (carrots), *Lactuca sativa* (lettuce), and *Allium cepa* (onion), supplying fruits and vegetables to Canada and the northeastern United States [16].

In 2016, a large-scale project was set-up in collaboration with local growers to develop conservation strategies adapted to their soils. As Bourdon et al. (2021) explained [11]: "In addition to controlling wind and, to a lesser extent, water erosion, a combination of other strategies is currently being investigated to extend the lifespan of these highly fertile lands including the application of plant biomass with a high carbon (C)/nitrogen (N) ratio (woodchip and grass straw)." Such strategies could potentially compensate for soil losses based on recent C modeling [12,17] and restore physical properties [18].

Miscanthus (*Miscanthus x giganteus*) straw and willow (*Salix miyabeana*) chips have been found to be good candidates for these conservation strategies [12,18,19], despite some anticipated side-effects such as N and P immobilization [11,20]. Miscanthus and short-rotation willow can be grown as biomass crops in degraded land [21,22] with little fertilization and practically no weed control [23,24], making them excellent choices for on-farm soil amendment production. Unlike manure and sewage sludge, for example, these plant-based amendments comply with food safety norms regarding vegetable cultivation and are not a source of pollution for rivers and bodies of water near the amended sites [25,26].

Approximately 14–20% of peatlands are used globally for agriculture, and when drained and cultivated, they represent some of the world's most productive agricultural soils [27]. Hence, the need for the development of sustainable management practice is urgent for these precious soils. The proposed conservation strategy is new and original as it aims to maintain long-term vegetable production on these productive lands while ensuring a massive return of organic matter annually, through biomass crop production on-site, or a biomass supply locally grown on more degraded surfaces.

The general objective of this field study was to assess the impacts of incorporating plant-based amendments, specifically *Miscanthus x giganteus* straw and *Salix miyabeana* chips, on nutrient availability for the roots of *Lactuca sativa* crops cultivated on peatland. We hypothesized that (1) both amendments similarly reduced the availability of nutrients, mainly nitrogen; that (2) the intensity of the reduction observed decreased over time; and that (3) rhizosphere strategies compensate for the effect of amendment.

## 2. Materials and Methods

### 2.1. Study Site

The soil at the research site (45°11′ N, 73°20′ W) is a moderately decomposed Haplosaprist [28], with a pH of 5.7 and a carbon and nitrogen content of 46% and 2.0%, respectively.

The average annual precipitation (30-year period) at the experimental site is 961 mm, and the annual average temperature is 6.6 °C. The annual frost-free period is 146 days, with 3289 degree-days above 0 °C (http://climate.weather.gc.ca/climate_normals/).

Although the field experiment was conducted over only one crop growth cycle, the experimental design was rigorously developed and applied, thus assuring the representativeness of the data, which are characteristic of this site and year, specifically.

### 2.2. Soil Amendment Description

Two biomass crops were selected in this experiment to amend the histosol: *Miscanthus x giganteus* (miscanthus) and *Salix miyabeana* (willow). These amendments were chosen based on the findings of Dessureault-Rompré et al. [12], who observed miscanthus and willow biomass material to be more resistant to degradation than the annual plant sorghum (*Sorghum bicolor*). The harvested biomass crops were bought from nearby producers. The characteristics of the two soil amendments are presented here in Table 1.

**Table 1.** Characteristics of the chopped plant material.

| Characteristics | Miscanthus | Willow |
| :---: | :---: | :---: |
| C (%) | 45.3 | 46.4 |
| N (%) | 0.198 | 0.343 |
| P (%) | 0.048 | 0.031 |
| C/N | 228 | 135 |
| C/P | 242 | 381 |
| Hemicellulose (%) | 29.8 | 10.9 |
| Cellulose (%) | 23.0 | 32.2 |
| Lignin (%) | 34.8 | 35.5 |
| Lignin/N | 176 | 103 |

### 2.3. Experimental Design

A 1 ha area of a 7 ha field was divided into three sections: (1) control, (2) miscanthus-amended, and (3) willow-amended. Each experimental section covered an area of about 330 m². Chopped miscanthus and willow material was added to the soil at a rate of 15 t ha$^{-1}$ (Figure 1). This rate is thought to be adequate to compensate for the average annual C losses observed in this type of soil (Dessureault-Rompré et al., 2020).

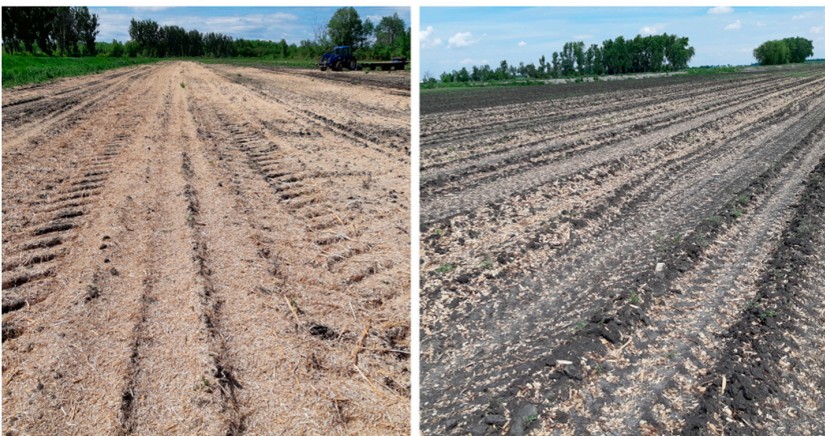

**Figure 1.** Experimental sections (miscanthus on the **left** and willow on the **right**) pictured before the chopped biomass was incorporated into the soil.

On 12 June 2019, the baled miscanthus was chopped and spread using a FP240 (New Holland Agriculture, New Holland, PA, USA) pull-type forage harvester equipped with a hay pickup. The bales were loosened and laid by hand in windrows on the ground to be picked up by the forage harvester and blown from the chute onto the plot. As the willow was already chipped when received, a lime spreader was used to spread it over the plot. The chopped plant material was then incorporated to a depth of 15 cm using a chisel.

### 2.4. Ionic Exchange Resin

The use of Plant Root Simulator (PRS®) ion exchange resins is an economical and rapid way to quantify the concentrations of a range of nutrients and contaminants in soils by simulating their uptake by plant roots [29–31]. In this study, the soil nutrient supply was assessed using PRS® cationic and anionic exchange resin probes. At six locations in each experimental plot, two cationic exchange resin probes and two anionic exchange resin probes were inserted at a depth of 15 cm in the root zone of two lettuces and into two root exclusion cylinders (bulk soil) measuring about 15 cm in height and 10 cm in diameter (Figure 2). The resin probes remained in the soil for seven days, and they were then replaced with fresh ones (Table 2). The removed resin probes were cleaned with a toothbrush and rinsed with demineralized water to remove any soil particles that could continue to exchange ions with the resins. Although laborious, this cleaning procedure is

essential to ensure that no soil particles are left on the resin surface, which could continue to exchange nutrients. Finally, the cleaned resin probes were stored at 4 °C before being sent to Western Ag Innovations' laboratories (Saskatoon, SK, Canada) for complete analysis. At Western Ag Innovations' laboratories, inorganic N (ammonium and nitrate) in the eluant was determined colorimetrically using automated flow injection analysis (Skalar San++ Analyzer, Skalar Inc., Breda, The Netherlands). The remaining nutrients (P, K, S, Ca, Mg, Fe, Mn, Cu, and Zn) were measured using inductively coupled plasma (ICP) spectrometry (Optima ICP-OES 8300, PerkinElmer Inc., Waltham, MA, USA).

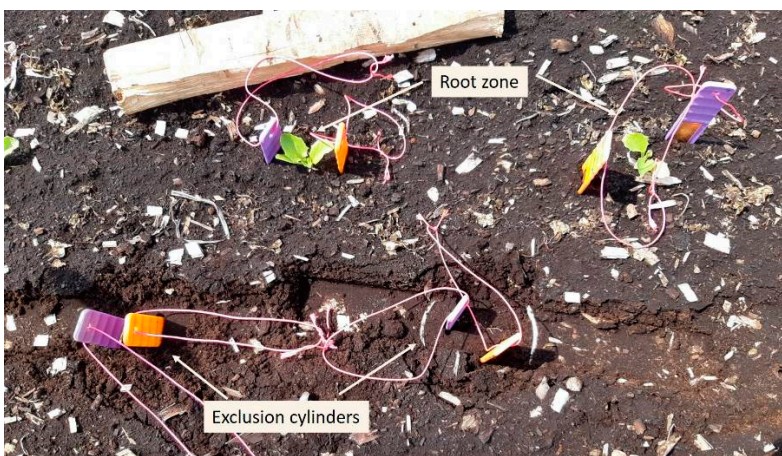

**Figure 2.** Experimental design with PRS® probes installed in the root zone and in root exclusion cylinders. The probes are shown here before they were completely inserted into the soil.

**Table 2.** Burial and retrieval dates for the PRS® probes and related periods.

| Burial Date | Retrieval Date | Period |
|---|---|---|
| 2019-07-10 | 2019-07-17 | 1 |
| 2019-07-17 | 2019-07-24 | 2 |
| 2019-07-24 | 2019-07-31 | 3 |
| 2019-07-31 | 2019-08-07 | 4 |
| 2019-08-07 | 2019-08-14 | 5 |
| 2019-08-14 | 2019-08-19 | 6 |

*2.5. Lettuce Crop Management and Yield Evaluation*

Small, five-day-old lettuce plants were transplanted on 1st July. All treatments were fertilized equally at planting with N-P-K (14.4-3.6-17.4) at a rate of 833 kg ha$^{-1}$. The plots were irrigated on 1st July (5 mm), 10th July (5 mm), and 6th August (15 mm). On 7th July, the first set of resin probes was installed in the field. Plot yields (fresh plant matter) were evaluated on 19th August, by weighing the lettuce harvested from one linear meter, repeated three times per treatment.

*2.6. Soil Sampling and Analysis*

Following the lettuce harvest, composite soil samples were taken at a depth of 0–20 cm at each of the locations previously occupied by the resin probes. The soil samples were then dried at 70 °C, sieved at 2 mm, and used for chemical analyses.

Nitrogen was extracted from the soil with water, using a 1:10 (*w/v*) soil:solution ratio. After an agitation period of 1 h at 200 rpm (19 mm circular orbit), the extract was centrifuged at 3000 rpm (2060× *g*) for 10 min and filtered through a No. 42 Whatman filter paper. Ammonium ($NH_4^+$) and nitrate ($NO_3^-$) concentrations were quantified with a Quikchem 8500 Series 2 system (Lachat Instruments, Loveland, CO, USA), using Quikchem methods 10-107-06-2-B and 12-107-04-1-F, respectively. Total N in the water extract was measured using the same system with prior persulfate oxidation, according to the procedure

described by Qualls (1989) [32]. Soluble organic N (SON) was calculated by subtracting $N-NO_3^-$ and $N-NH_4^+$ from total N.

Mehlich III elements were extracted with a 1 g of soil to 30 mL of solution ratio, a 5-min agitation period at 200 rpm (19 mm circular orbit), and filtration through a No. 42 Whatman filter paper [33]. Elements were analyzed by inductively coupled plasma–mass spectrometry (Icap 6500 MK2 radial, ThermoFisher Scientific, USA) within 12 h of extraction to prevent organic compounds from precipitating with metals.

For the active carbon fraction, a 0.02 M $KMnO_4$/0.1 M $CaCl_2$ solution was used according to the procedure described in Weil et al. (2003) and Blair et al. (1995) [34,35]. Total organic carbon and total nitrogen concentrations were determined using a LCN-2000 dry combustion analyzer (Leco Corporation, St. Joseph, MI, USA).

### 2.7. Data Visualization and Statistics

Data visualization was performed using the RStudio ggplot2 and emmeans packages. Analyses of variance with the *post hoc* Tukey HSD test were computed using the mixed procedure (nlme library) (RStudio Team, 2020). A natural logarithmic transformation was used when needed to respect the normality and homogeneity of variance assumptions.

## 3. Results

### 3.1. Yield (Fresh Mass)

Lettuce fresh mass decreased by 35% in the miscanthus-amended treatment and by 14% in the willow-amended treatment, although these differences were not statistically significant (Figure 3, Table 3).

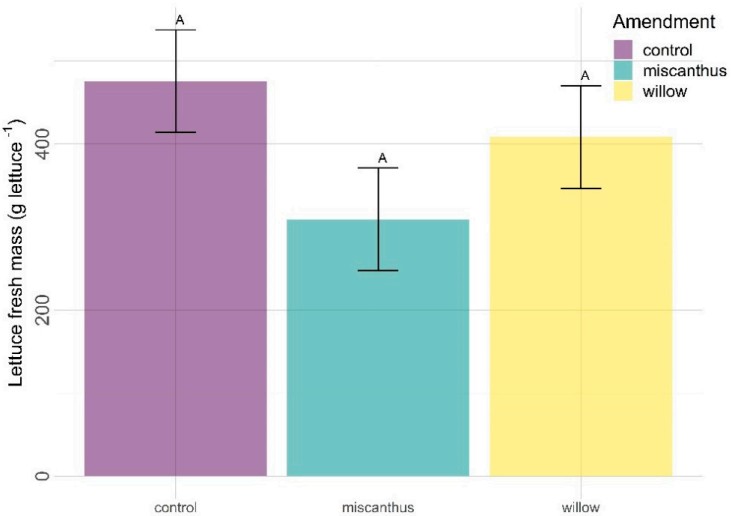

**Figure 3.** Lettuce fresh mass per plant as a function of soil amendment. Different letters indicate significance at $p < 0.05$ using the *post hoc* Tukey HSD test.

**Table 3.** Tukey's *p*-values for lettuce yield and all nutrients analyzed with the PRS®.

|  | Amendments | Sampling Zone | Time | Amendment X Sampling Zone | Amendment X Time | Sampling Zone X Time |
|---|---|---|---|---|---|---|
| Lettuce yield | 0.175 | NA | NA | NA | NA | NA |
| N | **<0.0001** | **0.0012** | **<0.0001** | **0.0134** | 0.9938 | 0.5338 |
| P | **0.0027** | 0.4923 | **0.0001** | 0.2515 | 0.4876 | 0.4075 |
| K | 0.4556 | 0.8435 | **<0.0001** | 0.1581 | 0.9557 | **0.5026** |
| Ca | 0.6404 | **<0.0001** | **<0.0001** | 0.3570 | **0.0148** | **0.0489** |
| Mg | 0.0906 | **<0.0001** | **<0.0001** | 0.8260 | **0.0165** | **0.0058** |
| S | **0.0005** | **0.0412** | **<0.0001** | 0.3465 | 0.7285 | **0.0173** |
| Fe | 0.4555 | **0.0002** | **<0.0001** | 0.1651 | 0.8025 | **<0.0001** |
| Mn | 0.3166 | **0.0039** | **<0.0001** | 0.0541 | 0.8456 | **0.0008** |
| Cu | 0.3731 | **0.0110** | **<0.0001** | 0.1131 | 0.9251 | 0.1421 |
| Zn | 0.9325 | **0.0275** | **<0.0001** | 0.2657 | 1.0000 | 0.1650 |

*3.2. PRS® Probes*

Overall, the addition of plant-based amendments significantly decreased the N and P supply rates and significantly increased the S supply rate (Table 3). Significant differences were observed between the sampling zones, with N, Ca, Mg, and S showing significant decreases and Fe, Mn, Cu, and Zn showing significant increases in the root zone as compared to the bulk soil.

The following sections present more detailed results for each of the aforementioned nutrients.

3.2.1. Nitrogen

N supply rates differed significantly with amendments, sampling zone, time, and interaction between sampling zone and time (Table 3). The N supply rate decreased significantly in both the miscanthus- ($-76\%$) and willow- ($-75\%$) amended treatments as compared to the control (Figures 4 and 5). Overall, the N supply rate decreased by 90% from the beginning to the end of lettuce growth and was 34% higher in the bulk soil than in the root zone. The significant interaction between amendment and sampling zone showed that the N supply rates were significantly lower in the root zone as compared to the bulk soil in the control and miscanthus-amended soil but not in the willow-amended soil (Figure 5). Finally, the plant N uptake index, calculated as the difference between the N supply rate in the bulk soil and that in the root zone (Figure 6), revealed a clear difference between the control treatment and the amended soils. While the plant N uptake index for the control treatment increased and then decreased markedly during lettuce growth, the index showed a consistent decrease in the miscanthus treatment for the same period. In the willow treatment, where the difference between the root zone and the bulk soil was not significant, the plant N uptake index was negative at the beginning of lettuce growth and increased slowly over time during lettuce growth.

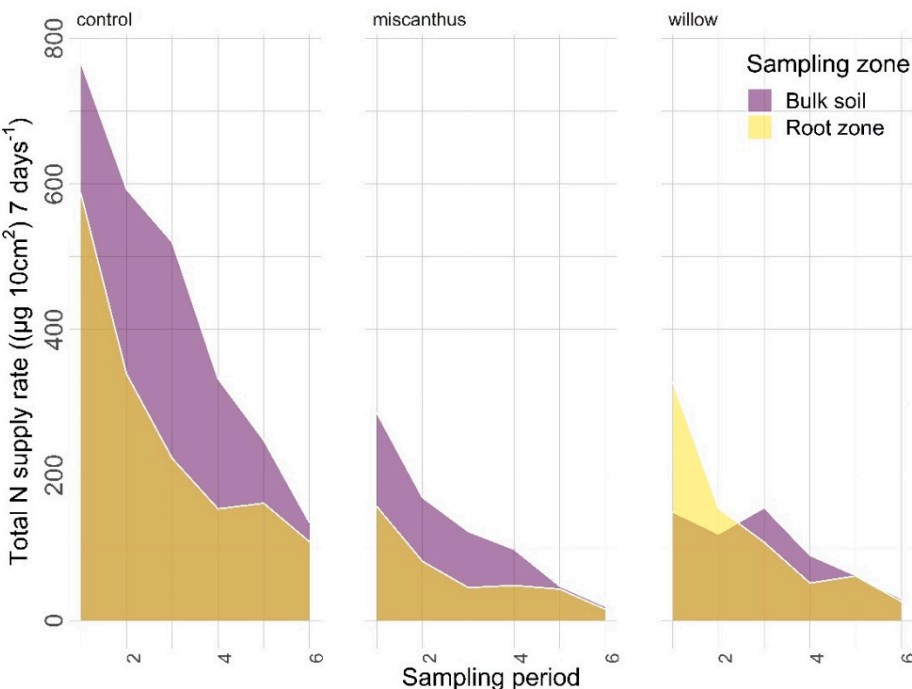

**Figure 4.** Total N supply rates measured in the bulk soil and the root zone as a function of time and amendment.

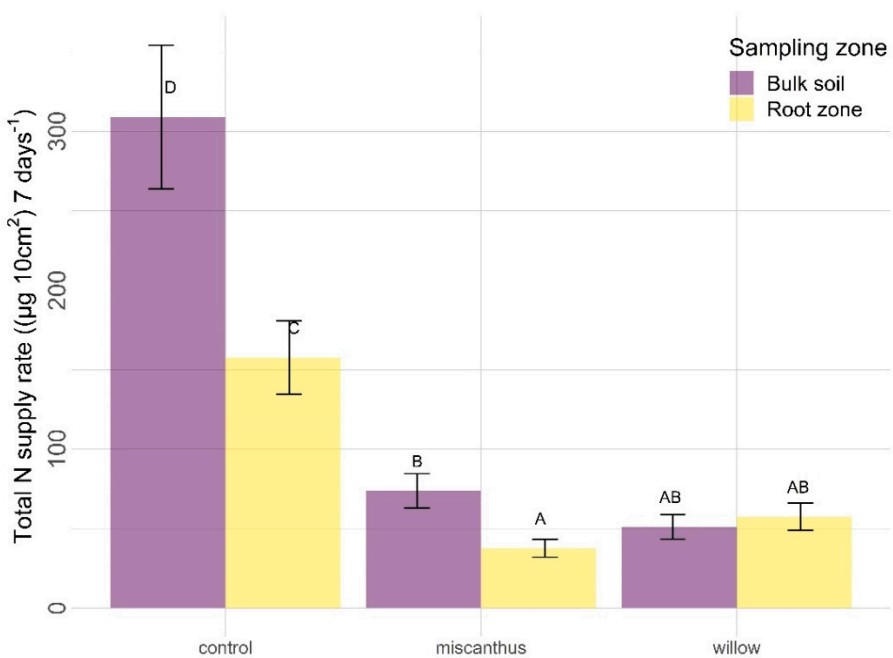

**Figure 5.** Statistical differences for total N supply rates between sampling zones and treatments. Different letters indicate significance at *p* < 0.05 using the *post hoc* Tukey HSD test.

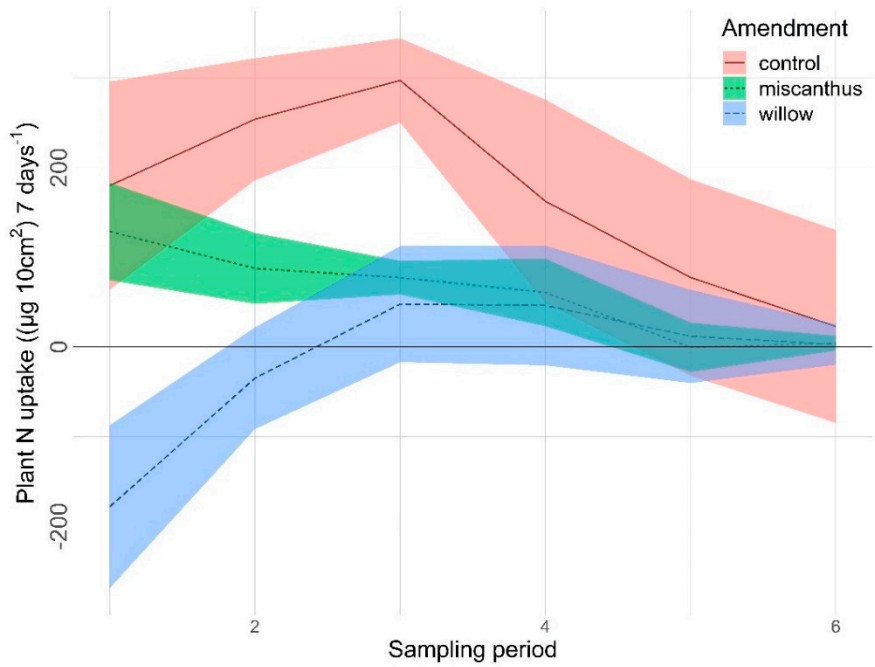

**Figure 6.** Plant N uptake index as a function of time and amendment. The colored area represents the error envelope as plotted with the geom_ribbon function in the ggplot2 R package.

### 3.2.2. Other Nutrients

*Phosphorus.* For P, amendments and time showed significant differences (Table 3). The P supply rate was significantly lower (−24%) in the willow treatment than in the miscanthus treatment and the control (Figure 7, left). In addition, the P supply rate varied over the lettuce growth period (Figure 7, right).

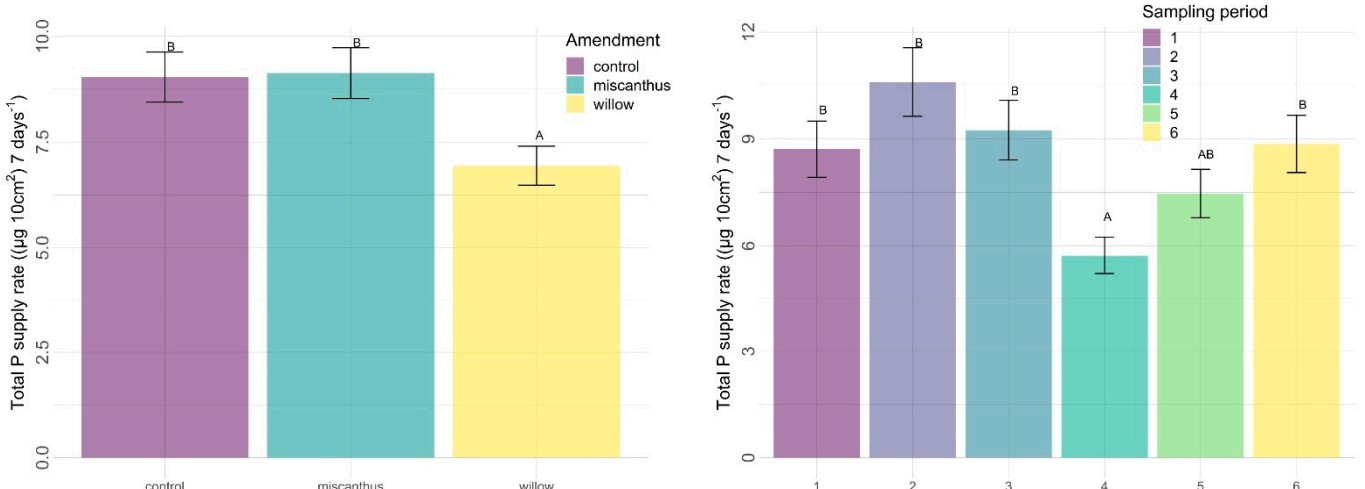

**Figure 7.** P supply rates as a function of amendment (**left**) and time (resin sampling period) (**right**). Different letters indicate significance at $p < 0.05$ using the *post hoc* Tukey HSD test.

*Potassium.* For K, only the PRS® probe sampling period showed a statistically significant difference, with a decrease over time (Table 3, Figure 8). From the beginning to the end of lettuce growth, K supply rates decreased by 30%.

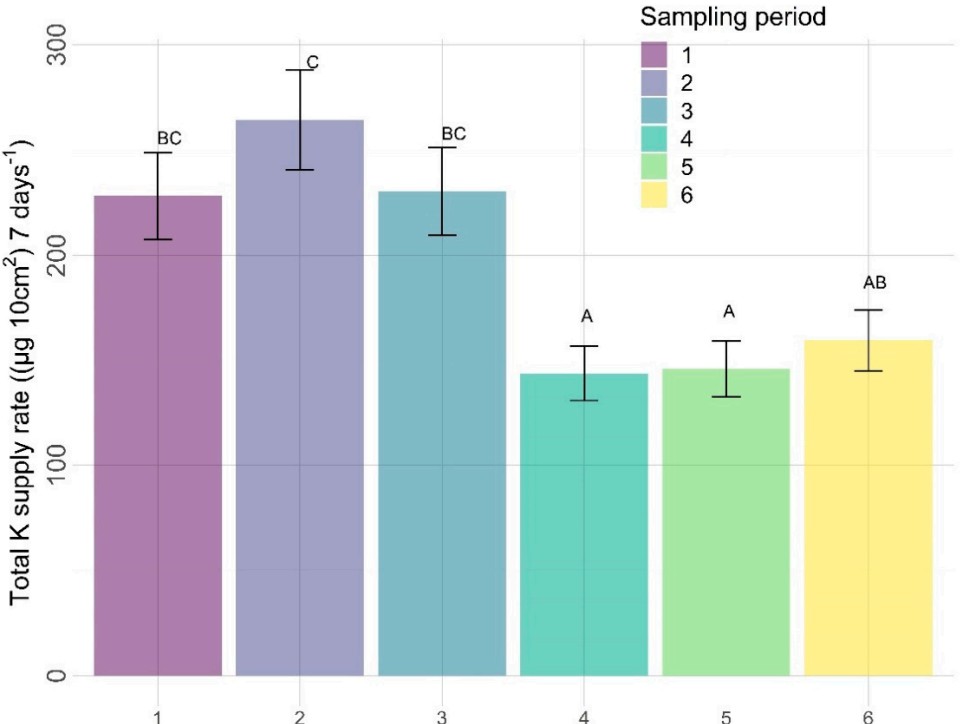

**Figure 8.** K supply rates as a function of time (resin sampling period). Different letters indicate significance at $p < 0.05$ using the *post hoc* Tukey HSD test.

*Calcium.* For Ca, sampling zone, time, amendment X time, and sampling zone X time all showed significant differences (Table 3). Although the amendment X time interaction was significant, it was not relevant from a practical point of view, as, for example, the control in the first sampling period differed from the miscanthus in the fifth sampling period. Ca supply rates decreased by an average of 30% across amendments and sampling zones from the beginning to the end of lettuce growth (Figure 9). In addition, the average difference between root zone and bulk soil sampling was 10%; the decrease in the root

zone was significant for resin sampling periods 3 and 4, with differences of 18% and 17%, respectively, between the root zone and the bulk soil (Figure 9).

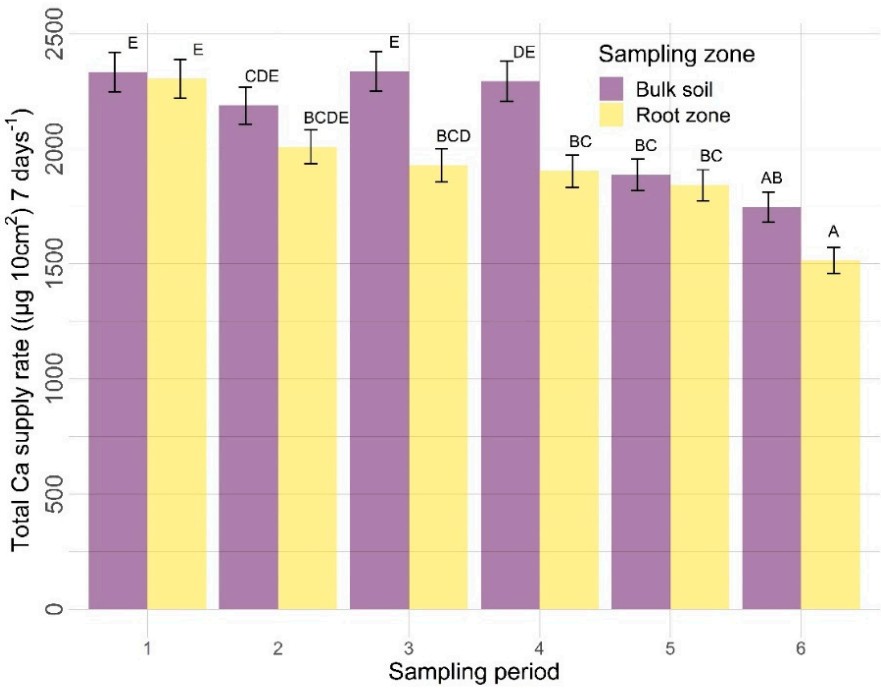

**Figure 9.** Ca supply rates as a function of sampling zone and time. Different letters indicate significance at $p < 0.05$ using the *post hoc* Tukey HSD test.

*Magnesium.* The behavior of Mg closely followed that of Ca. Sampling zone, time, amendment X time, and sampling zone X time all showed significant differences (Table 3). As mentioned above for Ca, the amendment X time interaction was significant but not relevant in practical terms. Mg supply rates decreased by an average of 18% across amendments and sampling zones by the end of lettuce growth (Figure 10). In addition, the average difference between root zone and bulk soil sampling was 12% and the decrease was significant for the third resin sampling period, with a 21% difference between the root zone and the bulk soil (Figure 10).

*Sulfur.* S supply rates differed significantly by amendment, sampling zone, time, and interaction between sampling zone and time (Table 3). The S supply rate increased significantly in both the miscanthus (+38%) and the willow (+59%) treatments (Figure 11, left). Overall, the S supply rate increased by 42% from the beginning to the end of lettuce growth and was 18% higher in the bulk soil than in the root zone. However, there was a significant interaction between the period and sampling zone, and the S supply rate was significantly higher in the bulk soil compared to the root zone in the fourth resin sampling period (Figure 11).

*Iron.* For Fe, sampling zone, time, and sampling zone X time all showed significant differences (Table 3). On average, Fe supply rates decreased by 43% across amendments and sampling zones from the beginning to the end of lettuce growth (Figure 12). In addition, the average difference between root and bulk soil sampling was 32%. The interaction between time and sampling zone showed that in the fourth period of resin sampling, the Fe supply rate was significantly higher (+244%) in the root zone as compared to the bulk soil (Figure 12).

*Manganese.* The behavior of Mn closely followed that of Fe. Sampling zone, time, and sampling zone X time all showed significant differences (Table 3). Mn supply rates had decreased by an average of 47% across amendments and sampling zones by the end of lettuce growth (Figure 13). In addition, the average difference between root zone and bulk soil sampling was 26%. The interaction between time and sampling zone showed

that in the fourth and fifth resin sampling periods, Mn supply rates in the root zone were, respectively, 125% and 93% higher than in the bulk soil (Figure 13).

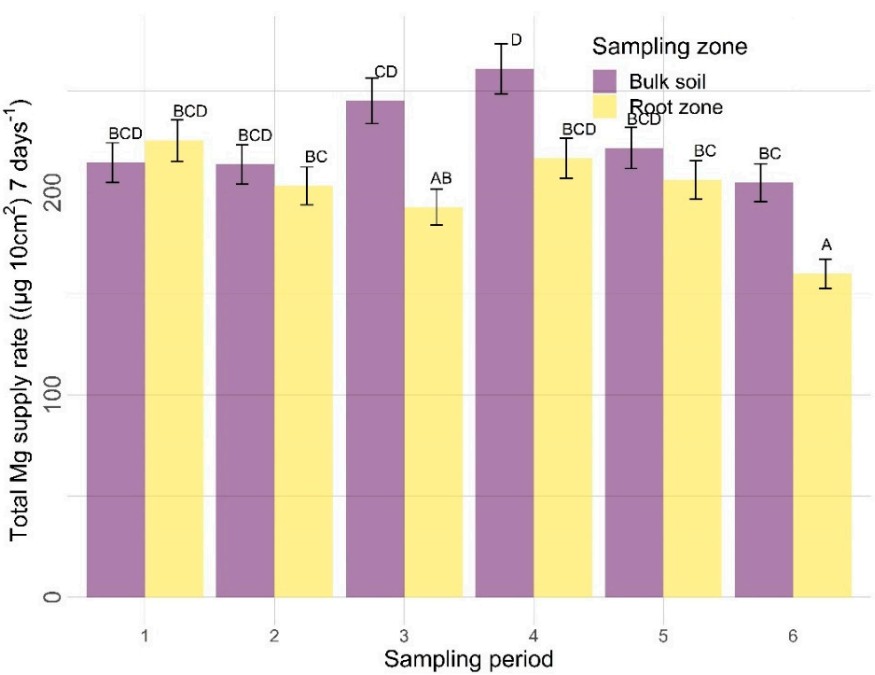

**Figure 10.** Mg supply rates as a function of sampling zone and time. Different letters indicate significance at *p* < 0.05 using the *post hoc* Tukey HSD test.

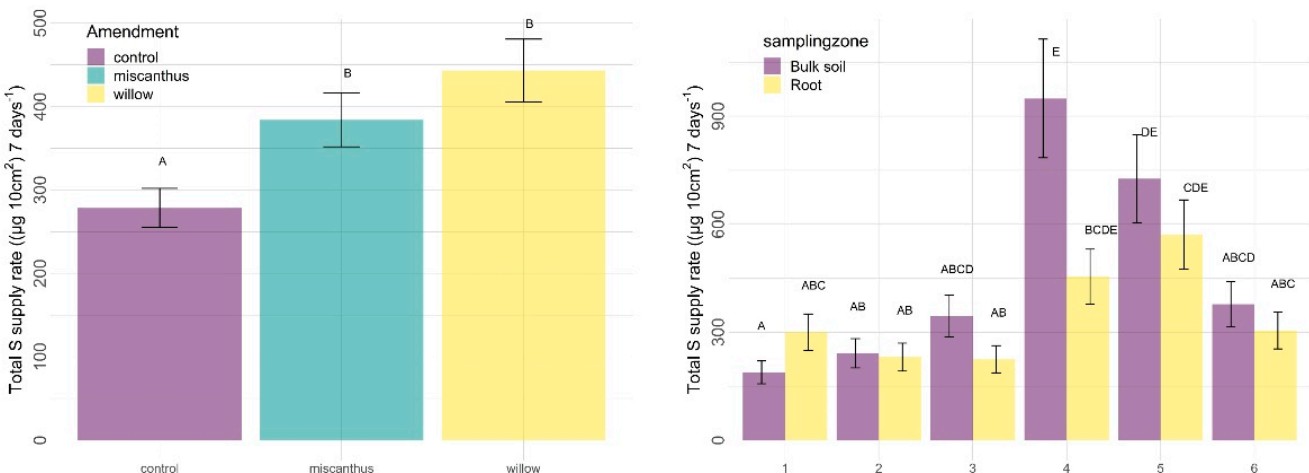

**Figure 11.** S supply rates as a function of amendment (**left**) and interaction between sampling zone and time (**right**). Different letters indicate significance at *p* < 0.05 using the *post hoc* Tukey HSD test.

*Copper.* Cu supply rates differed significantly by sampling zone and time (Table 3). Overall, Cu supply rates were 24% higher in the root zone than in the bulk soil (Figure 14, left). In addition, a significant decrease (50%) was observed between the beginning and the end of lettuce growth (Figure 14, right).

*Zinc.* Zn supply rates differed significantly by sampling zone and time (Table 3). Overall, Zn supply rates were 96% higher in the root zone than in the bulk soil (Figure 15, left). In addition, a significant decrease (61%) was observed between the beginning and the end of lettuce growth (Figure 15, right).

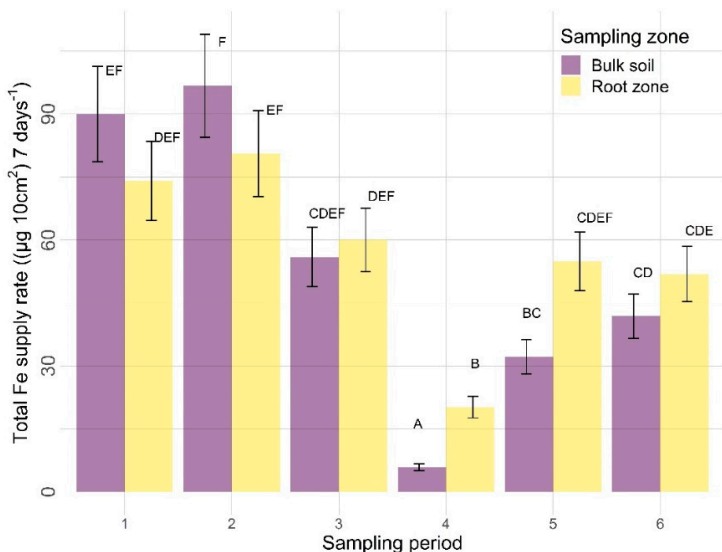

**Figure 12.** Fe supply rates as a function of sampling zone and time. Different letters indicate significance at *p* < 0.05 using the *post hoc* Tukey HSD test.

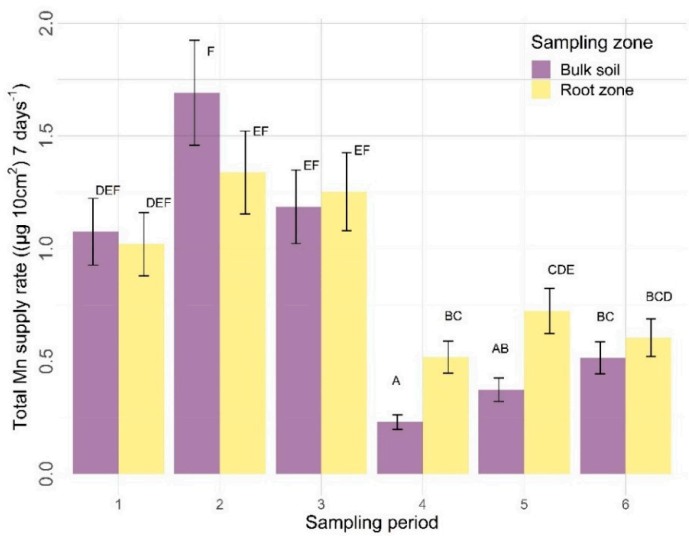

**Figure 13.** Mn supply rates as a function of sampling zone and time. Different letters indicate significance at *p* < 0.05 using the *post hoc* Tukey HSD test.

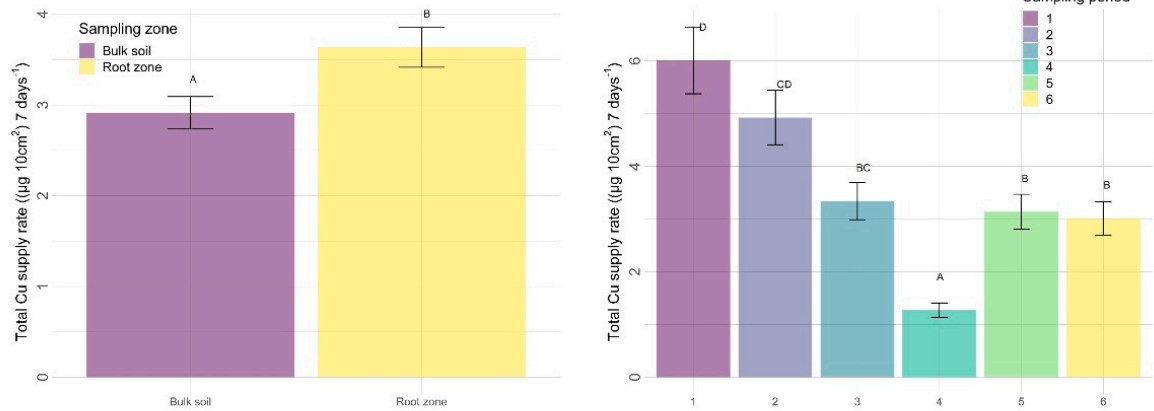

**Figure 14.** Cu supply rates as a function of sampling zone (**left**) and time (sampling period) (**right**). Different letters indicate significance at *p* < 0.05 using the *post hoc* Tukey HSD test.

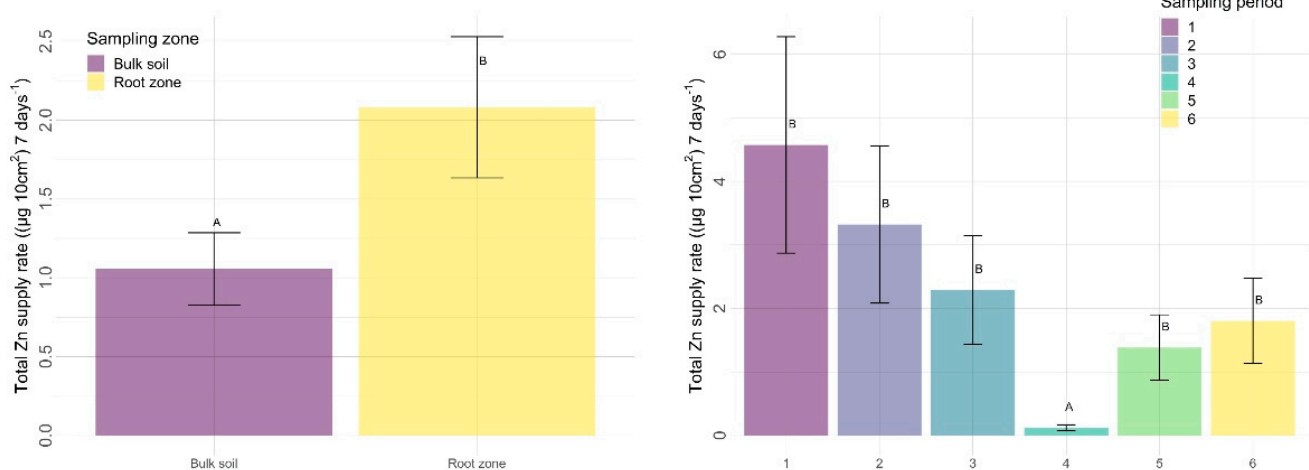

**Figure 15.** Zn supply rates as a function of sampling zone (**left**) and interaction between sampling zone and time (**right**). Different letters indicate significance at $p < 0.05$ using the *post hoc* Tukey HSD test.

### 3.3. Soil Samples following Lettuce Harvest

After harvesting the lettuce, soil samples (0–15 cm) were taken and analyzed (Table 4). The results showed that soluble organic N was the only parameter significantly influenced by the amendments, with significant decreases in both the miscanthus and the willow treatments. A tendency for Cu to be lower under willow treatment and for Fe to be higher under miscanthus treatment was also observed ($p = 0.055$).

**Table 4.** Mean $\pm$ (se) and *p*-values (Tukey's test) for different soil parameters measured in the soil at the end of lettuce growth as a function of amendment.

|                   | **Control**         | **Miscanthus**      | **Willow**           | ***p*-Values** |
|-------------------|---------------------|---------------------|----------------------|----------------|
| Mineral N         | 60.4 ± (7.5)        | 58.1 ± (10.0)       | 62.4 ± (6.2)         | 0.933          |
| Soluble organic N | **70.1 ± (9.6)**    | **21.0 ± (3.2)**    | **30.1 ± (6.2)**     | **0.005**      |
| Active C          | 10 604.0 ± (172.0)  | 10 779.0 ± (189.4)  | 12 430.7 ± (1487.2)  | 0.331          |
| C                 | 45.9 ± (0.4)        | 46.5 ± (0.2)        | 46.6 ± (0.4)         | 0.331          |
| N                 | 1.9 ± (0.0)         | 2.0 ± (0.0)         | 2.0 ± (0.0)          | 0.617          |
| C/N               | 23.7 ± (0.1)        | 23.9± (0.1)         | 23.5 ± (0.2)         | 0.164          |
| P                 | 314.7 ± (39.3)      | 219.7 ± (25.3)      | 201.7 ± (35.5)       | 0.110          |
| K                 | 1 91.2 ± (66.4)     | 324.3 ± (19.4)      | 328.0 ± (33.3)       | 0.120          |
| Ca                | 14 033.3 ± (550.7)  | 13 891.0 ± (497.6)  | 13 886.7 ± (456.6)   | 0.970          |
| Mg                | 1 107.3 ± (34.9)    | 1 135.0 ± (62.5)    | 1 138.3 ± (15.1)     | 0.970          |
| Fe                | 900.0 ± (40.3)      | 754.7 ± (9.9)       | 831.7 ± (38.9)       | 0.055          |
| Mn                | 16.3 ± (1.6)        | 17.4 ± (0.8)        | 21.5 ± (1.9)         | 0.106          |
| Cu                | 26.8 ± (3.4)        | 26.6 ± (1.7)        | 37.8 ± (3.2)         | 0.055          |
| Zn                | 22.2 ± (3.3)        | 27.3 ± (2.9)        | 34.6 ± (3.9)         | 0.105          |

## 4. Discussion

### 4.1. Nitrogen

The use of plant-based, high-C/N-ratio amendments has been proposed as a conservation strategy in cultivated peatland (Dessureault-Rompré et al., 2020). In the present study, PRS® probes were used to evaluate the impact of incorporating miscanthus straw and willow chip amendments in the field on nutrient availability in the rhizosphere of a lettuce crop. The field was fertilized as usual by the grower. Miscanthus and willow amendments decreased fresh lettuce yield by 35% and 14%, respectively. Although not statistically significant, such a yield decrease is necessarily relevant for growers as it could have substantial economic consequences.

Straw mulches from different plant sources have been studied and used in vegetable crop production for years, with both positive and negative effects on crop yield [36–38]. In the present study, however, the miscanthus straw and willow chips were not used as a mulch but were incorporated into the soil as an organic amendment. Few studies have investigated the effect of straw incorporation on vegetable crop yields. More numerous studies have focused on the effect of straw incorporation on wheat or wheat-rice rotation [39–41] and corn yields [42–44], with most reporting a neutral to positive crop response. In the present study, however, the quality of the straw was different, and straw was added at a much higher rate.

The present study highlighted the significant impact of miscanthus straw and willow chip incorporation on the field soil nutrient supply rate. Nitrogen was by far the most affected by the soil amendments, with a 75% reduction in the N supply rate. In a modeling study on the effect of 8 t ha$^{-1}$ of wheat straw incorporation on nitrogen dynamics, Garnier et al. [45] observed a 13% decrease in the net amount of nitrogen mineralized 13 months after the straw was incorporated. In an incubation experiment, both wheat straw and spruce sawdust added at a rate of 4.5 t C ha$^{-1}$ [46] were found to cause nitrogen immobilization, reaching a maximum level of 42 kg N ha$^{-1}$ and then decreasing to 8–15 kg N ha$^{-1}$ after a few weeks. In a microcosm study, rice straw incorporated at a rate of 10 t ha$^{-1}$ was found to immobilize 16–39% of applied N over the course of a 160 days incubation period [47]. In the present study, in the bulk soil, the difference between amended treatment and the nonamended soil did not attenuate over time (42 days) for willow and increased in the miscanthus-amended treatment. In the rhizosphere, the difference between the amended and nonamended treatments increased over time for both amendments (Figure 4). In a 56 days incubation study, Bourdon et al. [11] observed that the addition of 11 t ha$^{-1}$ of miscanthus straw and willow chips to a moderately decomposed histosol reduced mineral N from a KCl extract by 60% and 54%, respectively, overall, with the effect of these two amendments increasing over the course of the incubation period. In a 182 days incubation experiment, Marmier [20] observed that following miscanthus straw and willow chip amendments at a rate of 15 t ha$^{-1}$, N immobilization remained stable, increasing or decreasing over the course of the incubation period, depending on the type of histosol and amendment. Finally, the plant N uptake index in the control was found to increase until week 3 and then decrease until the end of lettuce growth. The plant N uptake index was much lower in the miscanthus treatment than in the control treatment and decreased from the beginning to the end of lettuce growth. The index for the willow treatment was initially negative and increased slightly until week 3. These results showed that miscanthus and willow amendments, by their negative impact on the supply rate of N, decrease the capacity of the lettuce crop to absorb N, which is consistent with the impact observed on lettuce yield. Interestingly, soil extract at the end of the growing season showed no difference in mineral N concentrations between the control and amended soil. However, a significant decrease in soluble organic N was observed. Dynamic changes in the fast-cycling mineral N pool might be better captured using PRS® probes than soil extract; however, future research should explore the time evolution between mineral and organic N pools under field-amended cultivated peatland.

### 4.2. Other Nutrients
#### 4.2.1. Amendments

For nutrients other than N, the effect of the amendments was significant only on P and S, with the willow treatment decreasing the phosphorous supply rate and both the willow and miscanthus treatments increasing the sulfur supply rate (Figures 7 and 11). Several authors have reported P immobilization with the incorporation of straw under different pedo-climatic conditions [48–50], while others have noted increased available P or P losses [51–53]. Bourdon et al. (2021) pointed out that because the P content of Quebec peat soils can be excessively high [54], P immobilization could help reduce eutrophication. Although Bourdon et al. (2021) found S immobilization to occur during a 56 days incubation

experiment with miscanthus and willow straw, the mineralization–immobilization of S has been shown to be related to the C:S ratio of the amendment [55]. In the present field study using PRS® probes, the S supply was found to increase substantially, by 38% (miscanthus) and 59% (willow). The S supply changed over time, however, increasing to a peak in the fourth sampling period and then decreasing, indicating that S was mineralized and then immobilized during lettuce growth.

### 4.2.2. Sampling Zone

Significant differences were found in the root zone for all nutrients analyzed, except P and K. The effect of sampling zone showed decreases in N, Ca, and Mg in the root zone as compared to the bulk soil zone. The opposite effect was noted for S, Fe, Mn, Cu, and Zn. Depletion and accumulation zones have often been observed in the rhizosphere [56]. Lettuce crops have shown important quantitative differences in root exudates depending on soil type, even under well-fertilized conditions [57]. Although we did not find studies on root exudates from lettuce grown in cultivated peatland, root exudation and activities are known to be intimately related to plant nutrition [58] and, therefore, might have contributed to the micro-nutrient mobilization observed in this study.

### 4.2.3. Time

Overall, the nutrient supply rate decreased over time, with few exceptions. Initial fertilization, lettuce uptake, and soil processes such as immobilization, mineralization, and adsorption to the soil matrix could all have been implicated to different degrees in the results obtained in the present study.

### 4.3. Retrospective on the Initial Hypotheses

The present study was developed based on three main hypotheses. The first one stated that both amendments similarly reduced the availability of nutrient, mainly nitrogen. This is actually true for nitrogen where the overall impact of miscanthus and willow on the reduction of the N supply rate was −76% for miscanthus and −75% for willow. However, more precise information came from the investigation of this effect at the root zone scale and allowed us to put forward that the supply of N under the willow amendment seemed slower for the first half of the lettuce growth period (see Figure 6) as compared to the N supply rate observed under the miscanthus treatment. In addition, few other nutrients were impacted by the amendment (only P and S).

The second hypothesis stated that the intensity of the reduction observed decreased over time. In the present study, we distinguished this effect for the bulk soil and root zone. In the bulk soil, the difference between the amended treatment and the nonamended soil did not attenuate over time for willow and increased in the miscanthus-amended treatment. In the rhizosphere, the difference between the amended and nonamended treatments increased over time for both amendments (see Figure 4).

The third and last hypothesis stated that rhizosphere strategies compensate for the effect of amendment. At the temporal (weekly) and spatial (root zone and bulk soil) scales used in this study, we did observe that in the rhizosphere zone under willow amendment, the N supply rate was higher as compared to the N supply rate observed in the bulk soil for the first two weeks and half of the lettuce growth, which could lead to a nutrient mobilization strategy (Figure 4). However, with the data in hand, we cannot distinguish between a mobilization strategy or a slower N uptake by the lettuce root. In addition, the difference between the root zone and the bulk soil under miscanthus treatment and for the control treatment was identical (−49% in the root zone as compared to the bulk soil).

## 5. Overall Perspective: Found and Missing Pieces of the Proposed Conservation Strategy Puzzle

The use of plant-based amendments with high C/N ratios as a conservation strategy in cultivated peatland is still in its infancy. This field study over one lettuce growth cycle ex-

amined the effects of such amendments in a real farming situation to evaluate the potential use of this conservation practice for sustainable vegetable crop production on cultivated peatlands. Although the pieces of the puzzle are starting to fit into place [11,12,19,20], this first field study revealed that some important information is still missing. Using PRS® probes under field conditions, this study confirmed that the main impact on crop growth resulting from the incorporation of miscanthus straw and willow chips is linked to N availability, with a consequent reduction in fresh lettuce yield. The plant N uptake index indicated how miscanthus and willow amendments can impact N uptake by the lettuce crop, pointing toward the need for fertilization adjustments under this conservation strategy.

As for the missing pieces, a number of unanswered questions remain with regard to optimizing the application of such amendments, such as the optimal time (fall versus spring) of application, the effects of combining this strategy with other conservation strategies such as cover crops (for example, a fall application of the amendment together with a cover crop), the effects of composting miscanthus straw and willow chips before incorporating them in the soil, and also the possibility of using these plant-based amendments as a mulch for the first year before incorporating them into the soil. Additional work is also needed to investigate the effects of consecutive years of soil amendment on different vegetable crops and in different types of cultivated peatlands to confirm and generalize the findings of the present study. Future field studies should also explore the long-term carbon dynamics in peatland amended with plant-based, high-C/N-ratio amendments under real field conditions to determine if this strategy could offset annual C losses and ensure muck soil conservation for generations to come.

**Author Contributions:** Conceptualization: J.D.-R.; methodology, J.D.-R. and A.G.; formal analysis, J.D.-R. and A.G.; investigation, A.G.; resources, J.D.-R. and J.C.; data curation, J.D.-R. and A.G.; writing—original draft preparation, J.D.-R. and A.G.; writing—review and editing, J.D.-R., A.G. and J.C.; visualization, J.D.-R.; supervision, J.D.-R. and J.C.; project administration, J.D.-R. and J.C.; funding acquisition, J.C. All authors have read and agreed to the published version of the manuscript.

**Funding:** We would like to acknowledge the financial support of the *Natural Sciences and Engineering Research Council of Canada (NSERC)* through a *Collaborative Research and Development Grant* in partnership with *Productions Horticoles Van Winden Inc.*, *Les Fermes Hotte et Van Winden Inc.*, *Maraîchers J.P.L. Guérin et fils Inc.*, *Delfland inc.*, *Vert Nature Inc.*, *Isabelle Inc.*, *La Production Barry Inc.*, *Le Potager Montréalais ltée.*, *Les Jardins A. Guérin et fils INC.*, *Les Fermes du Soleil Inc.*, *Les Fermes R.R. et fils inc.*, *Productions Maraîchères Breizh Inc.*, and *R. Pinsonneault et fils ltée.*

**Institutional Review Board Statement:** Not applicable.

**Acknowledgments:** We would like to acknowledge Moranne Béliveau, Charles Gauthier-Marcil, Simon Corbeil and Nicolas Shooner, who helped with the field study, particularly the cleaning of the resin probes.

**Conflicts of Interest:** The authors declare no conflict of interest.

**Sample Availability:** Samples of the compounds are not available from the authors.

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
