# Peer review of "Nutrient Availability for Lactuca sativa Cultivated in an Amended Peatland: An Ionic Exchange Study"

_nitrogen, doi:10.3390/nitrogen3010002_

Round 1
Reviewer 1 Report
Dear authors,
I rate the work highly, the manuscript requires minor corrections. I have included all my comments in the pdf file as the reviewer comments.
Please cite the suggested book:
Regards
Reviewer

Author Response
Response to reviewers
We appreciate the time and effort that you dedicated to providing feedback on our manuscript and are grateful for the insightful comments on and valuable improvements to our paper. We have incorporated most of the suggestions. Those changes are tracked within the manuscript.
Reviewer 1. All comments were included in a pdf file. The revised article submitted answers all the comments made by the reviewer, they are highlighted in yellow in the revised file. The axis titles in the figure were all changed according to reviewer comments.
Reviewer 2 Report
I would like to give recommendations on the work as follows:
- Authors should be more to emphasize manuscript novelty. Why is yours manuscript interesting for international readership?
- Lines 64-66 - The objective of this manuscript is too general. Authors should be more detailed described the aim of this research.
- Lines 73-74 – unclear is the number of degree-days, is it 3289 correct?
- Chapter Material and methods – Table 3 content interesting results of Tukey test and we can see that 10 different nutrients are analyzed. Chapter Material and methods should be content information which methods have been used for determination of P K Ca Mg S Fe Mn Cu.
- Table 3 – is it correct <.0001 or should it be <0.0001?
- Lines 490 and 557 – year of publication is missing.
- Line 455 – reference is incomplete.
- Chapter References - the names of journals are sometimes with capital letter and in the other case in lower case, compare for example lines 442 and 446. Please harmonize.
My recommendation is: accept manuscript after minor revision.
Author Response
Response to reviewers
We appreciate the time and effort that you dedicated to providing feedback on our manuscript and are grateful for the insightful comments on and valuable improvements to our paper. We have incorporated most of the suggestions. Those changes are tracked within the manuscript.
Reviewer 2. Please see below, in blue, for a point-by-point response to the reviewer 1 comments. The track-change mode in the submitted.
- Authors should be more to emphasize manuscript novelty. Why is yours manuscript interesting for international readership?
Thank you for this comment, we added at the end of the introduction the following sentence:
Approximately 14-20% of peatlands globally are used for agriculture and when drained and cultivated they represent some of the world’s most productive agricultural soils (IPS, 2008). Hence, the need for the development of sustainable soil management is urgent for these precious soils. The proposed conservation strategy is new and original since it aims to maintain long-term vegetable production on these productive lands while ensuring a massive return of organic matter annually, through biomass production on-site, or biomass supply locally.
- Lines 64-66 - The objective of this manuscript is too general. Authors should be more detailed described the aim of this research.
Thank you for this comment. In order to describe better the objective we added the following:
We hypothesized that (1) both amendments similarly reduced the availability of nutrient, mainly nitrogen, that (2) the intensity of the reduction observed decreased over time and that (3) rhizosphere strategies compensate for the effect of amendment.
In addition at the end of the discussion we added the following in order to follow up on the initial hypotheses:
The present study was developed based on three main hypotheses. The first one stated that (1) both amendments similarly reduced the availability of nutrient, mainly nitrogen. This is actually true for nitrogen where the overall impact of miscanthus and willow on the reduction of the N supply rate was -76% for miscanthus and -75% for willow. However more precise information came from the investigation of this effect at the root zone scale and allowed to put forward that the supply of N under the willow amendment seemed slower for the first half of the lettuce growth period (see Figure 6) as compared to the N supply rate observed under the miscanthus treatment. In addition few other nutrients were impacted by the amendment (P and S).
The second hypothesis stated that the intensity of the reduction observed decreased over time. In the present study, we distinguished this effect for bulk soil and root zone. In the bulk soil, the difference between amended treatment and the non-amended soil did not attenuate over time for willow and increased in the miscanthus-amended treatment, In the rhizosphere, the difference between the amended and non-amended treatments increased over time for both amendments (see Figure 4).
The third and last hypothesis stated that rhizosphere strategies compensate for the effect of amendment. At the temporal (weekly) and spatial (root zone and bulk soil) scales used in this study, we did observe that in the rhizosphere zone under willow amendment the N supply rate was higher as compared to the N supply rate observed in bulk soil for the first two weeks and a half of the lettuce growth which could lead to a nutrient mobilization strategy (Figure 4). However, with the data in hand we can’t distinguish between a mobilization strategy or a slower N uptake by the lettuce root. In addition, the difference between the root zone and the bulk soil under miscanthus treatment and for the control treatment was exactly the same (-49% in the root zone as compared to the bulk soil).
- Lines 73-74 – unclear is the number of degree-days, is it 3289 correct?
Yes, this is correct. The number of degree-days above 0°C at the experimental site is 3289.
- Chapter Material and methods – Table 3 content interesting results of Tukey test and we can see that 10 different nutrients are analyzed. Chapter Material and methods should be content information which methods have been used for determination of P K Ca Mg S Fe Mn Cu.
The following was added: Inorganic N (ammonium and nitrate) in the eluant is then determined colorimetrically using automated flow injection analysis (Skalar San++ Analyzer, Skalar Inc., Netherlands) The remaining nutrients (P, K, S, Ca, Mg, Fe, Mn, Cu, Zn) are measured using inductively coupled plasma (ICP) spectrometry (Optima ICP-OES 8300, PerkinElmer Inc., USA).
- Table 3 – is it correct <.0001 or should it be <0.0001?
Yes <0.0001is right. The correction was made as suggested.
- Lines 490 and 557 – year of publication is missing.
Year of publication was added.
- Line 455 – reference is incomplete.
The complete citation was added.
- Chapter References - the names of journals are sometimes with capital letter and in the other case in lower case, compare for example lines 442 and 446. Please harmonize.
The references were thoroughly revised.